# Self-powered and speed-adjustable sensor for abyssal ocean current measurements based on triboelectric nanogenerators

Yuan Chao Pan[1,2], Zhuhang Dai ®[1], Haoxiang Ma[1], Jinrong Zheng[1], Jing Leng[1], Chao Xie[1], Yapeng Yuan[1], Wencai Yang[1], Yaxiaer Yalikun[3], Xuemei Song[2], Chang Bao Han[2] ✉, Chenjing Shang[1,4] ✉ & Yang Yang ®[1] ✉

The monitoring of currents in the abyssal ocean is an essential foundation of deep-sea research. The state-of-the-art current meter has limitations such as the requirement of a power supply for signal transduction, low pressure resistance, and a narrow measurement range. Here, we report a fully integrated, self-powered, highly sensitive deep-sea current measurement system in which the ultra-sensitive triboelectric nanogenerator harvests ocean current energy for the self-powered sensing of tiny current motions down to 0.02 m/s. Through an unconventional magnetic coupling structure, the system withstands immense hydrostatic pressure exceeding 45 MPa. A variable-spacing structure broadens the measuring range to 0.02–6.69 m/s, which is 67% wider than that of commercial alternatives. The system successfully operates at a depth of 4531 m in the South China Sea, demonstrating the record-deep operations of triboelectric nanogenerator-based sensors in deep-sea environments. Our results show promise for sustainable ocean current monitoring with higher spatiotemporal resolution.

Deep ocean exploration and exploitation are essential to mapping the Earth's internal tectonic and evolutionary history, searching for marine minerals, obtaining evidence for Earth science, and interpreting trends in deep-sea ecosystems[1–3]. Metrics of current are essential components of deep-sea oceanographic parameters and are important to understanding changes in the seabed geological structure, trends in the global climate, and the distribution of biological communities[4–6]. However, exploring the deep ocean, especially to the abyssal zone (at depths of 4000–6000 m), presents a complex challenge to the designers and manufacturers of instrumentation for deep-sea deployment, primarily owing to the extreme environment of the abyssal ocean. Furthermore, the tiny ocean flow (typically on the order of centimeters per second) means that sensors need to be ultra-sensitive. At present, mainly electromagnetic, acoustic, and

mechanical instrumentation is used for ocean current monitoring[7]. The measurement accuracy of an electromagnetic current meter is reduced by external magnetic field interference. Although the acoustic current meter has the highest precision and can perform profiling measurements[8], it is unsuitable for turbid or ultra-clean water and highly reflective near-bottom environments[9]. Moreover, its cost hinders its widespread use.

The mechanical current meter has the advantages of a simple structure, affordable price, and no blind measurement area. Despite its potential to be widely deployed for distributed ocean flow field observation, the conventional mechanical current meter has limited pressure resistance and a narrow measurement range. Moreover, all the aforementioned ocean current meters require an external power supply (e.g., batteries) for signal transduction. Limited battery power

[1]Institute of Deep-Sea Science and Engineering, Chinese Academy of Sciences, Sanya, China. [2]The Key Laboratory of Advanced Functional Materials, Ministry of Education of China, Faculty of Materials and Manufacturing, Beijing University of Technology, Beijing, China. [3]Division of Materials Science, Nara Institute of Science and Technology, Nara, Japan. [4]Shenzhen Key Laboratory of Marine Bioresource and Eco-environmental Science, College of Life Sciences and Oceanography, Shenzhen University, Shenzhen, China. ✉e-mail: cbhan@bjut.edu.cn; cjshang@szu.edu.cn; yangyang@idsse.ac.cn

inevitably leads to the under-sampling of ocean states. Observations may not reflect actual trends at any given location and time because so few measurements are collected. These problems limit the potential of state-of-the-art current meters for long-term, large-scale, and distributed ocean observations with high spatiotemporal resolution and long duration.

Distributed renewable energy sources provide an impetus for developing distributed sensing systems[10]. Triboelectric nanogenerators (TENGs) are cost effective[11,12], low weight[13–15], and highly efficient at harvesting low-frequency mechanical energy[16,17] and have been applied as micro and nano energy sources[18–21], blue energy harvesters[22–26], and self-powered sensors[27–30]. For self-powered sensors, the transformation from mechanical signals to recognizable electrical sensing signals does not need an additional power supply, and the TENG can perform this function well. For example, TENG-based self-powered sensing has been extensively studied in speed, vibration, biomedicine, and other sensing fields[31–35]. Recently, TENGs have demonstrated great potential in the development of self-powered sensors for ocean observations[36–38]. In fluid dynamics sensing, TENGs have been widely studied for wind speed[39–41], pipeline velocity[42,43], and raindrop sensing[44]. However, research on the use of TENGs in ocean current sensing in the deep sea is still lacking. Realizing a highly sensitive design that is suitable for monitoring subtle ocean current flow and that has a simple and effective structure capable of withstanding a high-pressure, highly corrosive deep-sea environment is a challenging task. Completing this task is important to building a low-cost deep-sea current observation network and promoting the deep-sea application of TENGs.

Herein, we report a fully integrated, self-powered, high-pressure-resistant, and ultra-sensitive ocean current monitoring system (DS-TENG) capable of measuring flows at depths down to 4531 m. The system employs a flexible TENG with biological fur and flexible foils for current energy harvesting and sensing, which reduces the friction resistance of the structure and effectively increases the sensitivity and durability of the structure. Through a magnetic coupling design, the DS-TENG achieves the highly efficient non-contact transmission of ocean current energy, generating a flow-velocity-dependent electrical signal (without the need for an external power supply). Notably, we further design a variable-spacing mechanism, which effectively modifies the distance of the magnetic coupling and adjusts the speed measurement range. The measurement range of the DS-TENG is 0.02–6.69 m/s, which is 67% wider than that of the traditional vertical-axis current sensor. Furthermore, this paper reports a field experiment conducted on the DS-TENG at depths down to 4531 m in the South China Sea. To the best of our knowledge, this is the first demonstration of a TENG-based sensor in the abyssal ocean. The DS-TENG operated continuously for 6 h at a flow velocity of 0.16 to 0.76 m/s with a linear correlation between the output signal and flow velocity up to $R^2 = 0.97$. The results show that the DS-TENG provides a low-cost and reliable sensing capability for deep-sea velocity observations and has excellent potential in the construction of future ocean stereoscopic observation networks.

## Results

### Structures and working principle of the DS-TENG
The development of sustainable and efficient ocean energy harvesting technology is important to the construction of a distributed ocean observation network. Here, we design a TENG-based ocean current sensor for energy harvesting and self-powered ocean current measurements.

Figure 1a shows the exploded view of the DS-TENG, which mainly comprises a pressure-resistant and waterproof sealing and tank; printed circuit board for data acquisition, processing, and communication; TENG and magnetically coupled and variable-spacing structure; and rotating cup structure for energy harvesting and ocean current

sensing. The rotating cup structure comprises six conical cups symmetrically arranged around the center of a rotating disk. When the sensor is exposed to ocean currents, the differential water pressure (relating to the varying shapes of the cups) on the two sides of the rotating axis rotates the cups. The harvested rotational energy is transferred to the TENG via the transmission structure. Here, we adopt a contactless magnetic coupling structure instead of a standard contact structure (Fig. 1b), as the former is conducive to the design of a sealed pressure-resistant structure for the harsh deep-sea environment.

We also propose a variable-spacing structure (Fig. 1c) for expanding the measurement range. By configuring the convexity in chute 1 (state 1) or chute 2 (state 2), two magnetic-coupling transmission clearances corresponding to different velocity measurement ranges, are realized. The variable-spacing structure means that the system has a wider measurement range and better pressure resistance than commercial current meters and can thus measure flow in a deep-sea high-pressure environment.

Finally, a highly flexible TENG (Fig. 1d) is designed with low-friction-resistance flexible fluorinated ethylene propylene (FEP) and fur as tribomaterials. Even for tiny ocean currents in the deep-sea environment, the rotor rotates smoothly and harvests current energy with a linear correlation between the current speed and frequency of the output electrical signal.

The working mechanism of the TENG in converting ocean current energy into electrical energy is shown in Fig. 1e and is based on the triboelectrification and electrostatic induction effect[26]. First, the fur and FEP film are electrically uncharged. When the fur and FEP film rub against each other, they generate an electrostatic charge through triboelectrification. Specifically, when the fur comes into contact with the FEP film, the triboelectric effect causes the fur and FEP film to take on equal amounts of positive and negative charge. Owing to the different contact areas (i.e., the area of the fur is half that of the FEP film), the $Cu^+$ and $Cu^-$ electrodes generate equal amounts of positive and negative charges through electrostatic induction (Fig. 1e–I). As the fur slides counterclockwise on the FEP to balance the non-mobile triboelectric charges in the FEP layer, the negative charge is transferred from the $Cu^-$ electrode to the $Cu^+$ electrode through the external circuit (Fig. 1e–II), generating a current from bottom to top. When the fur entirely coincides with the $Cu^+$ electrode, all the negative charge on the $Cu^-$ electrode is transferred to the $Cu^+$ electrode (Fig. 1e–III). Similarly, as the fur continues to rotate, the negative charge on the $Cu^+$ electrode returns to the $Cu^-$ electrode (Fig. 1e–IV), producing a current in the external circuit in the opposite direction. According to previous work[45], the formulas for calculating the open-circuit voltage ($V_{OC}$), short-circuit current ($I_{SC}$), and transferred charge ($Q$) are as follows:

$$Q = \int_0^l \frac{\sigma\omega dK}{1 + \left[\frac{c_2(k)}{c_1(k)}\right]_{x=g+l}} - \int_0^l \frac{\sigma\omega dk}{1 + \left[\frac{c_2(k)}{c_1(k)}\right]_{x=0}} \tag{1}$$

$$V_{OC} = \frac{Q}{C}$$

$$I_{SC} = \frac{dQ}{dt}$$

Here, $\sigma$ is the tribo-charge density, $\omega$ is the width of the fur, $dk$ is a small region at the bottom of the fur containing tribo-charges, and the total charges on the $Cu^+$ and $Cu^-$ electrodes are thus $\sigma\omega dk$. $c_i(k)$ is the capacitance between the small region $dk$ and the metal electrode.

The distribution of electric potential in the fur and FEP film at different stages (Fig. 1f) is simulated using COMSOL (COMSOL 5.6, COMSOL Inc., USA). The initial triboelectric charge density on the surfaces of the fur and FEP film is set at ±70 μC/m², and the gap

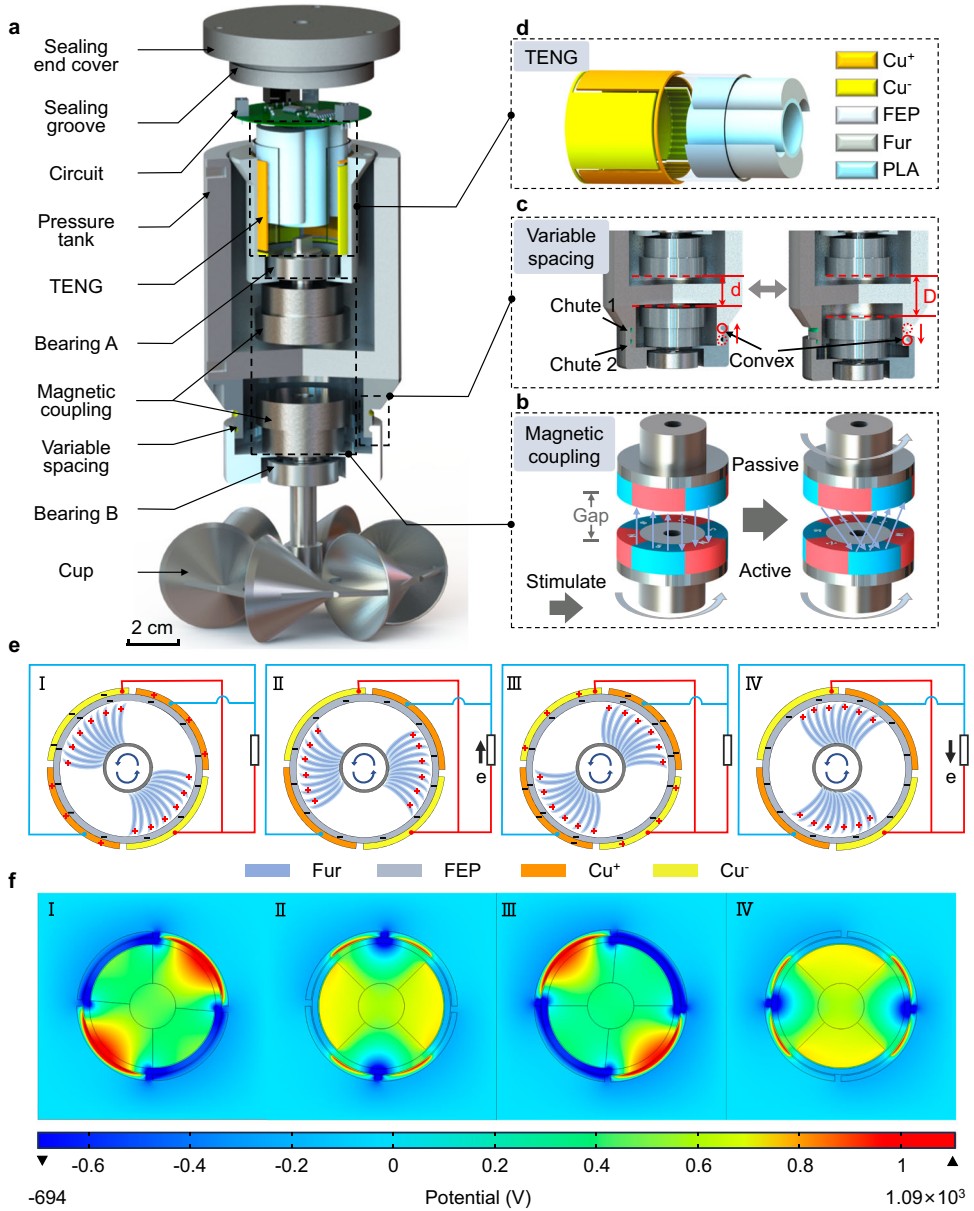

**Fig. 1 | Design and working principle of the DS-TENG. a** Exploded view of the TENG-based fully integrated, self-powered, and ultra-sensitive ocean current sensor. Three-dimensional structural schematics of the magnetic coupling (**b**), variable spacing (**c**), and TENG (**d**). **e** Schematic of the working principle of the DS-TENG. **f** Finite element simulation of the potential distributions in four states of the DS-TENG.

between the two surfaces is set at 1 mm. The potential difference is a maximum when the fur is directly above the $Cu^+/Cu^-$ electrodes. The results of the potential distribution are consistent with theoretical predictions.

## Material and structural optimization of the DS-TENG

To maximize the sensitivity, measurement range, and output stability of the DS-TENG, the materials and structure of the DS-TENG are optimized. This includes optimization of the treatment and length of the fur, selection, and thickness of the negative tribomaterials, number of electrodes of the TENG, and structure of the magnetic coupling and variable-spacing structure. The preparation of the fur as the positive tribomaterial for the TENG is shown in Fig. 2a. The surface of the fur is roughened by treatment with biological enzymes and plasma etching to create a greater contact area[46], which is conducive to generating more electrical charge during the friction process.

The effect of the fur length on the $V_{OC}$ of the TENG is shown in Fig. 2b. At a rotational speed of 100 rpm, $V_{OC}$ first increases and then decreases with an increase in the fur length. A maximum value of 121.6 V is reached when the fur length is 16 mm. The results are mainly due to the fur length affecting the effective contact area (Supplementary Fig. 1). When the fur is too short, it fails to make sufficient contact with the FEP surface. When the fur is too long, the fur makes contact with the positive and negative electrodes simultaneously, resulting in the cancellation of the induced potential at the outer part of the electrode pair. Moreover, longer fur leads to increased frictional resistance and, thus, reduced sensor sensitivity. The optimal fur length of 16 mm was thus obtained.

The effects of the materials and thickness of the negative tribomaterial on the output performance of the TENG are next investigated. Here, polyimide (PI), polytetrafluoroethylene (PTFE), and FEP as negative tribomaterials are tested. As depicted in Fig. 2c, the FEP-Cu combination has the best output performance. Furthermore, the

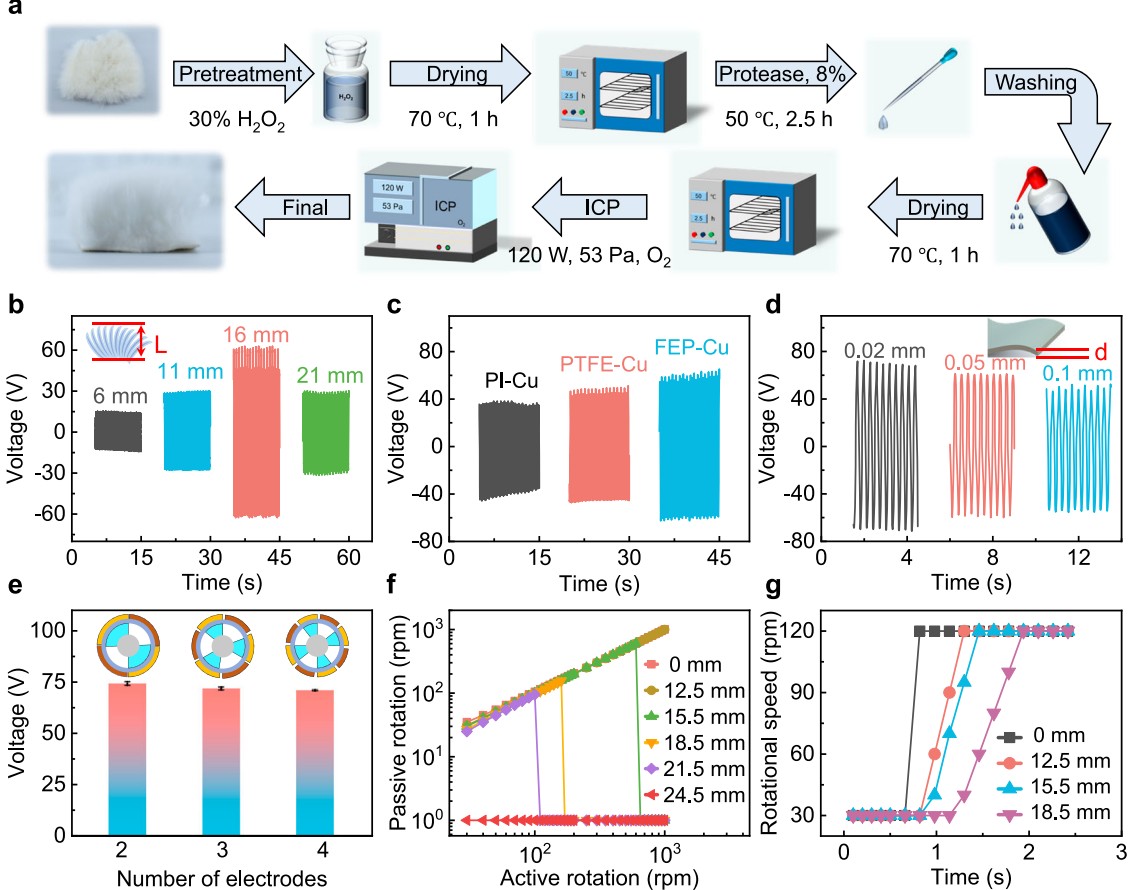

**Fig. 2 | Material and structural optimizations of the DS-TENG. a** Preparation of the fur as the positive tribomaterial for the TENG. The output voltage of the DS-TENG: (**b**) with different fur lengths, (**c**) with different tribomaterials, (**d**) with different thicknesses of fluorinated ethylene propylene, (**e**) with different numbers of electrodes. Error bars represent the standard deviation over three independent measurements. **f** Transmission curve and (**g**) transmission hysteresis curve of the active and passive magnets at different transmission clearances.

voltage decreases as the thickness of the FEP film increases from 0.02 to 0.1 mm (Fig. 2d), indicating that a thinner triboelectric layer results in more induced charge during the friction process. However, a 0.02-mm film is prone to damage during use, inconvenient for integration, and not suited to long-term use. A 0.05- mm FEP film is thus adopted as the dielectric layer.

The effect of the number of electrodes on the output performance is shown in Fig. 2e. The output voltage decreases slightly with an increase in the number of electrode pairs, mainly because the area of a single electrode decreases as the number of electrodes increases. The area of the fur, thus needs to be reduced with an increase in the number of electrodes. Furthermore, a greater number of electrodes results in the fur touching the positive and negative electrodes simultaneously, which reduces the output slightly. Finally, the long-term stability of the designed TENG is tested. After 30 days of operation, the reduction in the induced charge of the TENG remains within 6% (Supplementary Fig. 2). The cause of the slight reduction in TENG performance is mainly attributed to the material wear or contact deformation of the rabbit fur. Nevertheless, the deformation of the rabbit fur and the attenuation of the induced charge will slow down, and the output of the TENG tends to fluctuate within a certain range. In addition, after more than 30 days of use, the surface of the rabbit fur shows no appreciable wear or tear (where detailed analysis and SEM charts are presented in Supplementary Fig. 3), demonstrating the excellent long-term stability of the TENG.

Magnetic coupling realizes non-contact energy transmission through the interaction of the north and south poles of active and passive magnets, and the transmission clearance directly affects the design of the pressure-resistant tank and the maximum velocity measurement range. The effect of the transmission clearance (i.e., 0, 12.5, 15.5, 18.5, 21.5, or 24.5 mm) of the magnetic transmission structure on the maximum effective transmission rotational speed is shown in Fig. 2f. An increase in the transmission clearance reduces the maximum effective transmission rotational speed. When the clearance is less than 12.5 mm, the magnetic coupling drives continuously and stably in the speed region of 1000 rpm. When the clearance is 15.5, 18.5, and 21.5 mm, the maximum rotational speed of the passive magnet that can be achieved before transmission failure is 600, 160, and 100 rpm, respectively. When the clearance exceeds 24.5 mm, the magnetic coupling fails, irrespective of the rotational speed of the passive magnet. Detailed transmission curves for different transmission clearances are presented in Supplementary Fig. 4.

To investigate the synchronicity of the active and passive magnets in the transmission process, we record the response time of the passive magnet when the rotational speed of the active magnet is set from 30 to 120 rpm using a tachometer. The results (Fig. 2g) show an inevitable delay for the passive magnet due to the transmission clearance, with the response time increasing with the clearance. As the clearance increases from 12.5 to 18.5 mm, the response time increases from 0.48 to 0.96 s. Such a response time is perfectly adequate for measuring upper-ocean flow (which varies over the course of hours or days)[47] or abyssal flow (which varies over the course of months or seasons)[3].

In summary, the starting torque and magnetic coupling are negatively correlated to the magnetic transmission clearance within

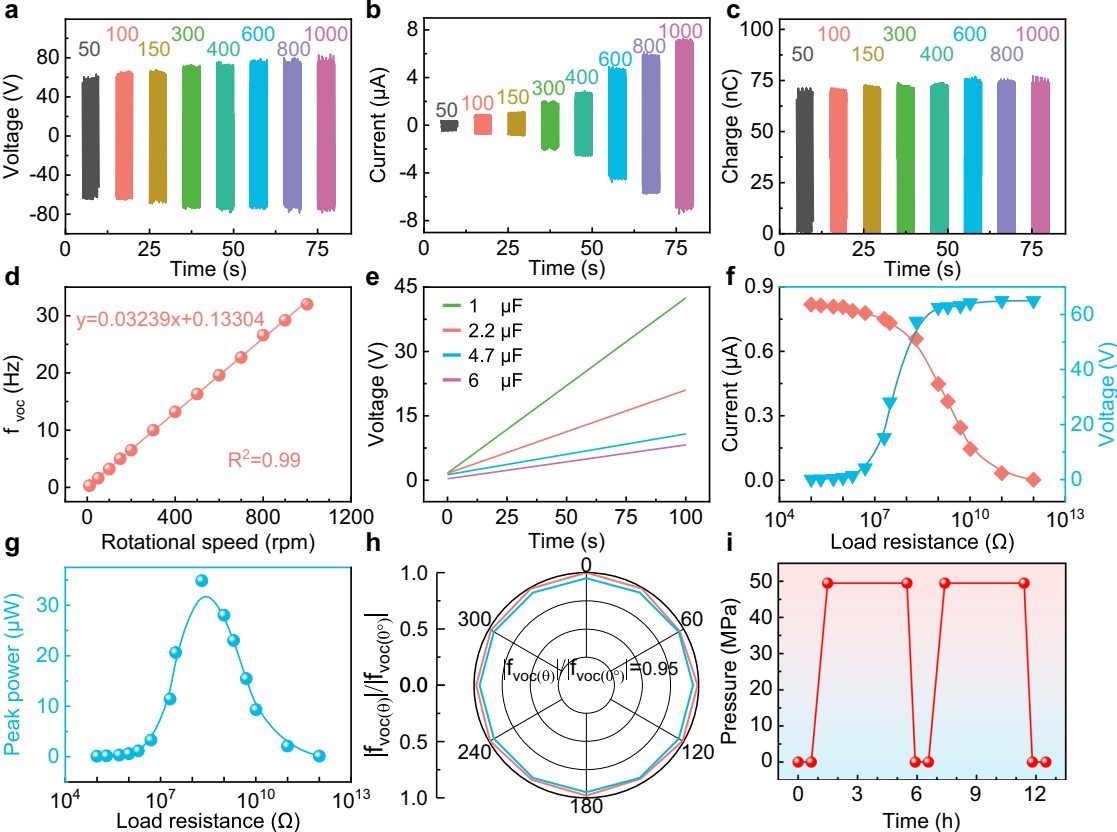

**Fig. 3 | Output performance and characterization of the DS-TENG. a** Open-circuit voltage, (**b**) short-circuit current, and (**c**) transferred charge of the DS-TENG at different rotational speeds of the motor (units: rpm). **d** Linear correlation between the output frequency of the voltage ($f_{voc}$) and motor speed of the DS-TENG. **e** Charging curves of different capacitors. **f** Resistance dependency of the output current and voltage of the DS-TENG. **g** Average output powers for different load resistances. **h** Polar plot showing the effect of the flow direction on the output performance of the DS-TENG. **i** A pressure cycle curve of the DS-TENG.

the effective transmission range. Challenges associated with the development of the DS-TENG include realizing a broad measurement range with sensitivity to low flow velocities and robustness against high flow velocities. We thus design a variable-spacing magnetic-coupling structure that has a large transmission clearance (18.5 mm) at a low flow velocity to ensure the sensitivity of the ocean current sensor during startup, and that has a small transmission clearance (12.5 mm) at a high flow velocity to ensure the strength of the vertical rotor and the stability of the rotation process (where more detailed explanations and diagrams are presented in Supplementary Fig. 5).

**Performance characterization of the DS-TENG**

To characterize the performance of the DS-TENG at different current speeds and different ocean depths, the output performance of the DS-TENG with a transmission clearance of 12.5 mm is tested using a stepper motor in the laboratory. The output voltage, short-circuit current, and transferred charge of the DS-TENG under different rotational speeds are shown in Fig. 3a–c. The DS-TENG experiences slight increases in both the output voltage and transferred charge with rotational speed, which are attributed to the enhanced centrifugal effect produced by the fur as the rotational speed continues to rise. This leads to closer contact between the fur and dielectric layer and, thus, an increase in the friction charge and induced charge and, ultimately, a slight increase in the output. The peak current is positively correlated with the rate of charge transfer. As the rotational speed increases, the frequency at which the fur slides on the positive and negative electrodes increases, which accelerates the transfer of the triboelectric charge in the external circuit, such that the peak current clearly corresponds with the rotational speed. At a rotational speed of

1000 rpm, $V_{p-p}$ reaches 157.61 V, $I_{p-p}$ reaches 14.35 μA, and the transfer charge reaches 74.17 nC. When the rotational speed is 2000 rpm, $V_{p-p}$, $I_{p-p}$, and the transferred charge reach 161.69 V, 19.25 μA, and 74.47 nC, respectively, as shown in Supplementary Fig. 6. Figure 3d shows the relationship between the speed of the stepper motor and the output frequency of the voltage ($f_{voc}$) of the DS-TENG at a transmission clearance of 12.5 mm. In the range of 0–1000 rpm, the relationship between the speed of the motor and $f_{voc}$ shows excellent linearity ($R^2 = 0.99$). This indicates that the signal frequency of the DS-TENG has excellent linearity with the external trigger frequency and can be used as an ideal sensing signal.

For simulating the operating conditions of deep-sea exploration equipment at low speed, the charging capacity of the DS-TENG is tested at a lower speed of 120 rpm, as shown in Fig. 3e. The DS-TENG charges a 6 μF capacitor to 8.1 V within 100 s. We next investigate the load current and voltage of the DS-TENG under different resistance conditions, as shown in Fig. 3f. We find that the load current of the DS-TENG decreases with an increase in the external load resistance, whereas the load voltage of the DS-TENG system increases with the external load resistance. Figure 3g shows the peak power of the DS-TENG system as a function of the external load resistance at 120 rpm. It is seen that a peak power of 34.86 μW is generated when the external load resistance is $2 \times 10^8$ Ω. Furthermore, we investigate the effect of the flow direction on the output performance of the DS-TENG (Fig. 3h). In the horizontal direction, for the same velocity stimulation from different directions, the ratio between $f_{voc}$ in other directions and f in the initial direction (0°) remains above 95%. This indicates that the DS-TENG maintains accurate and stable output performance for water motions from all directions. Furthermore, the pressure-resistance test

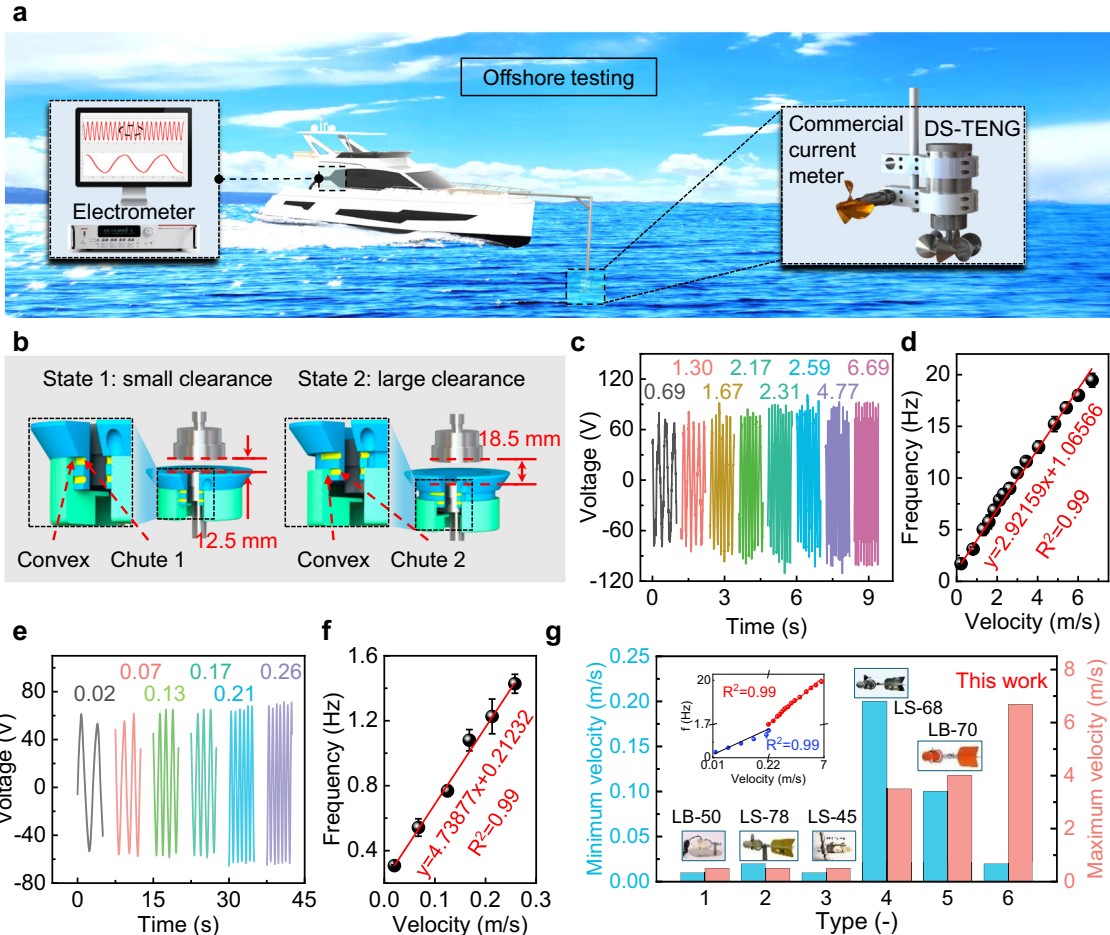

**Fig. 4 | Application of the DS-TENG in monitoring the flow velocity. a** Schematic of the offshore test platform. **b** Schematic of the variable-spacing structure. **c** Real-time voltage signals at high velocities (unit: m/s) measured in state 1. **d** Linear relationship between the voltage frequency and flow velocity of the DS-TENG in state 1. Error bars represent the standard deviation over three independent measurements. **e** Real-time voltage signals at low flow velocities (unit: m/s) measured in state 2. **f** Linear relationship between the voltage frequency and flow velocity of the DS-TENG in state 2. Error bars represent the standard deviation over three independent measurements. **g** Comparison of the flow velocity measuring range between the DS-TENG and commercial current meters.

results of the DS-TENG in a pressure chamber are shown in Fig. 3i. The sensor remains intact as the applied pressure is increased to 49.5 MPa, indicating that the DS-TENG can withstand deep-sea hydrostatic pressure to a depth of at least 4500 m.

### Flow velocity sensing of the DS-TENG

According to the working principle of the DS-TENG, a voltage is generated only when the fur inside the device slides against the surface of the FEP under the action of the ocean current, and the voltage is used as the velocity-sensing signal. As the peak current may be affected by the external environment (for example, the rotating speed), a more stable voltage frequency is chosen as the sensing signal. On this basis, a linear correlation between the DS-TENG's output electrical signal and the velocity can be established to monitor the velocity.

To test the DS-TENG's output performance in the natural ocean environment, we conducted a field experiment in the waters near Sanya Bay, China. As shown in Fig. 4a, calibration is carried out using the commonly applied still water tower method[48]. The DS-TENG and a commercial current meter are fixed side by side at a depth of 1 m in front of a speedboat. This arrangement places both instruments in a relatively consistent flow environment, which is convenient for testing.

A DS-TENG with two transmission clearances is designed to characterize the effect of the transmission clearance on the speed measurement range. For the small transmission clearance in state 1 (Fig. 4b), the magnetic transmission structure shows stronger coupling

and transmission stability, leading to better performance under the condition of a higher flow velocity (Fig. 4c and Supplementary Movie 1). The DS-TENG stably generates voltages at a flow velocity between 0.22 and 6.69 m/s with a linear correlation between the frequency of the voltage (f) (or rotating cup speed) and relative velocity as high as 0.99 (Fig. 4d and Supplementary Fig. 7a).

Similarly, for the large clearance in state 2, the magnetic transmission structure has a lower starting torque and is more sensitive to the low flow velocity, having a starting velocity of 0.02 m/s. Within the flow velocity range of 0.02 to 0.26 m/s (Supplementary Movie 2), the coefficient of correlation between f (or the rotating cup speed) and the relative velocity reaches 0.99 (Fig. 4e, f and Supplementary Fig. 7b). Combining the measurement capabilities of the two transmission clearances, the measurement range of the DS-TENG becomes 0.02 to 6.69 m/s, which is 67% wider than that of the traditional commercial current sensor (Fig. 4g). Such a wide measurement range makes the system useful across a wide range of applications, demonstrating the superiority of our design.

### Deep-sea applications of the DS-TENG

On the basis of the excellent performance of the DS-TENG in flow velocity sensing, we further integrate the DS-TENG on a remotely operated vehicle (ROV) to evaluate its operational performance in the harsh environment of the deep sea, especially that of the abyssal zone. The main differences in water between laboratory conditions and the

deep sea are the ambient pressure and density, which are factors that directly affect sensor performance. To offer a comprehensive view of our system in the deep sea, we present a schematic illustration and photograph of the ROV in operation (Fig. 5a, b). Closer images of the DS-TENG on the ROV are presented in Supplementary Fig. 8.

A signal processing system (Fig. 5c) is integrated within the DS-TENG to enable real-time signal transmission from the deep sea. Under the stimulation of ocean current, the DS-TENG generates voltages related to the flow velocity, which are then filtered and amplified through a signal processing circuit. The analog data are then converted into digital format and transmitted to a ship-based platform through an optical communication cable via the RS-232 protocol. The ship-based platform equipped with a data acquisition card reads and processes the data. Finally, the flow velocity is analyzed and visualized in real-time through a self-designed graphical user interface (where detailed circuit diagrams and analysis are presented in Supplementary Fig. 9).

A photograph and movie of the DS-TENG operating at a depth of 4531 m in the abyssal zone of the South China Sea (110°46′ E and 16°73′ N) are captured by a waterproof, deep-sea camera installed on the ROV (Fig. 5d, Supplementary Movies 3 and 4). As the ROV travels at different speeds, the onboard Doppler velocity log (DVL) records the relative flow velocity between the ROV and seawater in real-time. Meanwhile, the velocity data recorded by the DS-TENG are transmitted to the surface platform, ensuring a constant flow of real-time information (Supplementary Fig. 10). As shown in Fig. 5e, the DS-TENG effectively captures subtle changes in the flow velocity from 0.33 to 0.50 and 0.55 m/s. These changes are further evidenced by the corresponding waveform frequencies (Fig. 5f–h) of the voltages of 1.12, 1.22, and 1.31 Hz, respectively. In addition, this robust response to small velocity changes under abyssal zone conditions is supported by the linear correlation between the different relative flow velocities (between the ROV and seawater) and the output frequency of the DS-TENG (Fig. 5i). The output frequency of the DS-TENG shows good linear dependence ($R^2 = 0.97$) on the flow velocity from 0.16 to 0.76 m/s. Furthermore, validation tests conducted under various depth conditions confirm that the velocity measured by the DS-TENG aligns closely with the data from the DVL (Fig. 5j), and the DS-TENG thus has the potential not only to make sea current velocity measurements but also to determine the speed of submersibles, expanding its utility beyond conventional applications. Impressively, as demonstrated in Fig. 5k, after 240 min of continuous operation, both the DS-TENG and DVL consistently deliver results with an accuracy exceeding 88.5%, indicating that the DS-TENG has excellent stability. In addition, the maximum depth in the field experiment is 4531 m (Supplementary Movie 5). Throughout the field expedition, the surface platform accurately receives data transmitted by the DS-TENG, and the equipment remains fully operational after recovery, which verifies the technical feasibility of the DS-TENG velocity measurement in actual high-pressure environments of the abyssal ocean. This marks an advance in the ability of a TENG to monitor ocean currents in the challenging domain of the abyssal ocean.

## Discussion

In this paper, we presented the design and build of a TENG-based, self-powered deep-sea sensor for in situ continuous measurements of ocean current metrics. First, the vertical-axis rotating cup structure enables the DS-TENG to sense the two-dimensional omnidirectional flow velocity. The sensors feature a design of magnetic transmission and a variable-spacing structure in contrast with the previous contact design, improving the reliability of the DS-TENG in deep-sea environments and broadening the measurement range of the DS-TENG to 0.02–6.69 m/s, which is 67% wider than the measurement range of the traditional vertical-axis current sensor. Finally, we developed a complete signal processing and transmission system and successfully conducted an underwater sensing experiment at depths down to

4531 m. This represents the first application of TENG-based sensors at abyssal ocean depths, confirming the superior structural design and practical potential of the DS-TENG system in extreme environments. This work thus provides a new and effective method of addressing the limitations of the traditional current sensor, eliminating the need for an external power supply, and expanding the measuring range beyond the constraints of the rotating shaft design.

## Methods

### Fabrication of the TENG

The TENG comprises a rotor on the inside and a stator on the outside, with both components coaxially installed. The outer wall of the cylindrical rotor is symmetrically mounted with two brushes made of rabbit fur, each having a brush area of 18 mm × 37 mm. The inner wall of the stator is mounted with four pieces of Cu film, having dimensions of 15 mm × 37 mm × 0.065 mm. There is a gap of 2.5 mm between adjacent Cu films. FEP film with dimensions of 113 mm × 40 mm × 0.05 mm is adhered to the surface of the Cu film. The outside of the FEP film is attached to the Cu film, whereas the inside is in contact with the brush. The brush is coaxially installed with the magnetic coupling and transmits power through the rotating shaft.

### Fabrication of the ocean current sensor

The structure of the ocean current sensor mainly includes a rotating cup, magnetic transmission, and variable-spacing structure, the TENG, a printed circuit board, and a sealing tank. The structural design of the DS-TENG is modeled using mechanical engineering design software and initially validated for three-dimensional printed objects using a three-dimensional printer (CR5060 Pro, CREALITY, China). Finally, we use 316 stainless steel with equal parameters to replace the exposed parts. The pressure tank is made of 316 stainless steel, which is a high-strength austenitic stainless steel with high yield strength (310 MPa) and tensile strength (620 MPa). Therefore, this material is beneficial in terms of reducing the thickness of the isolation sleeve while meeting the requirements of deep-sea pressure resistance. The sealing end cover is designed on top of the DS-TENG, and the two sealing grooves between the end cover and pressure tank are sealed by an O-ring made of nitrile material, which is a commonly used sealing material. As the mechanical current meter cup has a mature commercial design and usage, we opted to adhere directly to the national standard of rotating-element current meters (GB/T11826-2019, China) in our design process. Six evenly distributed rotating cups are three-dimensionally printed from aluminum alloy. Each cup has a radius of 20 mm. The details of the DS-TENG rotating cup are presented in Supplementary Fig. 11. The transmission shaft is made from aluminum alloy and has a diameter of 8 mm. The shaft mainly acts as a coaxial connection between the rotating cups and the active magnet. The magnetic coupling is a standard part (model HSF06-D35-P6-d8S, HSF Magnets Ltd., China). The outer diameter, inner hole diameter, and height of the coupling are 35, 8, and 20 mm, respectively. The passive magnet is installed coaxially with the TENG rotor. The variable-spacing structure mainly comprises two rotating bodies. Rotating body 1 has an inner diameter of 22.75 mm and an outer diameter of 29 mm. The body has a semicircular chute with a diameter of 3 mm as an axial opening along its outer surface, and two semicircular chutes with diameters of 2.5 mm as radial openings along its outer surface. Rotating body 2 has an inner diameter of 29.5 mm and an outer diameter of 36 mm. The body has a convex protrusion with a diameter of 2.5 mm along the radial direction of its inner surface, which slides in the chute of rotating body 1. Rotating body 2 is fixed relative to the active magnet, whereas rotating body 1 is fixed relative to the passive magnet. The sealing tank system mainly comprises a sealing tank, lid, and O-ring seal. The inner diameter, outer diameter, and bottom thickness of the sealing tank are 54, 78, and 11.5 mm, respectively. The sealing lid has a diameter and thickness of 78 and 20 mm, respectively, and axial and radial sealing

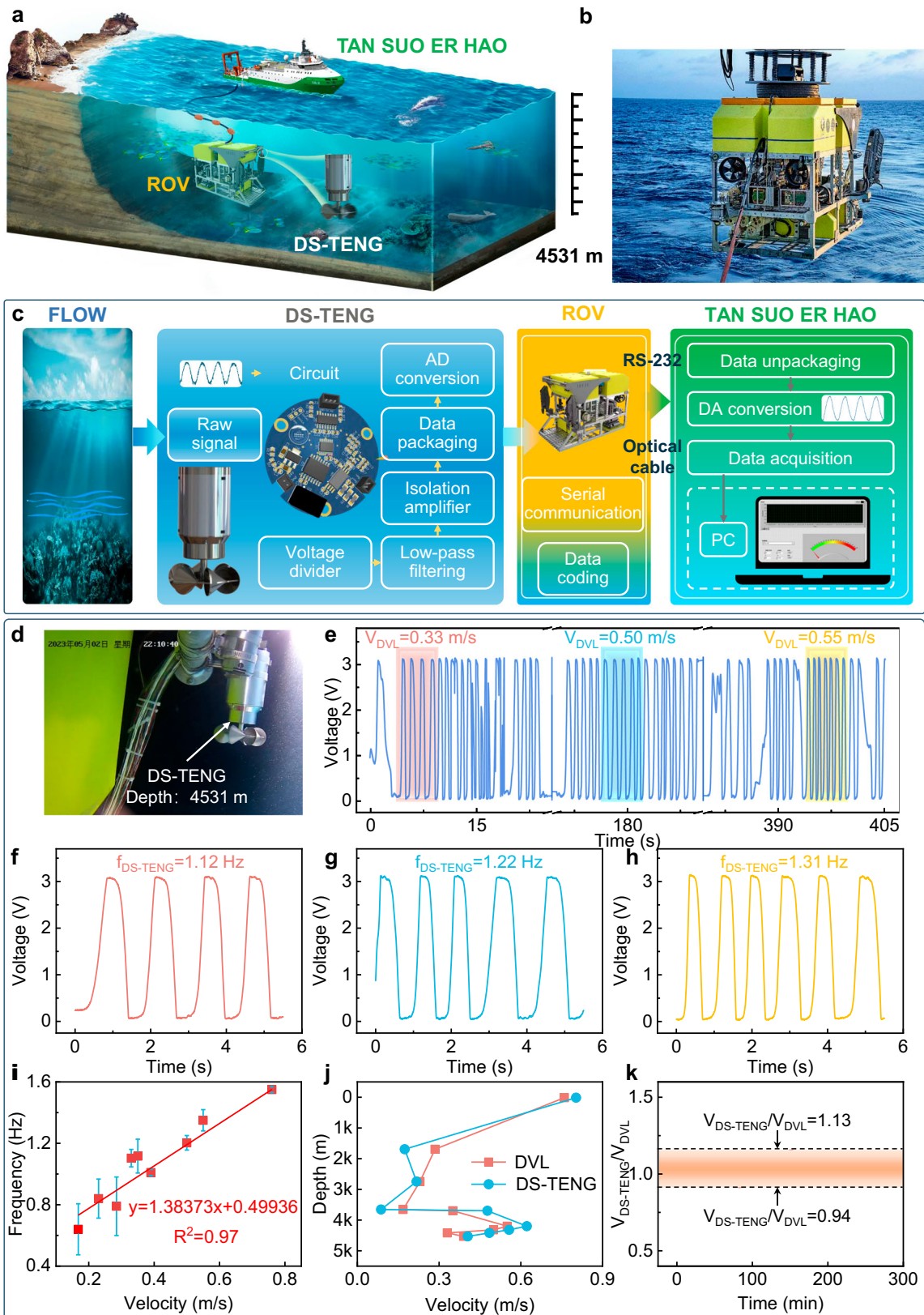

**Fig. 5 | Abyssal ocean-current measurements made using the DS-TENG.**
**a** Schematic of the deep-sea experiments and application of the DS-TENG. **b** Image of the DS-TENG integrated with a remotely operated vehicle (ROV). **c** Flowchart of the collection and processing of flow velocity monitoring data. **d** Photograph of the DS-TENG taken at a depth of 4531 m. **e** Raw data obtained by the DS-TENG on the ROV at different speeds. Voltage waveform of the DS-TENG with an ROV speed relative to sea current of 0.33 m/s (**f**), 0.50 m/s (**g**), and 0.55 m/s (**h**). **i** Linear relationship between the relative flow velocity and voltage frequency of the DS-TENG. Error bars represent the standard deviation over three independent measurements. **j** Depth profile of the flow velocity measured by the DS-TENG and a commercial Doppler velocity log. **k** Signal correlation between the DS-TENG and Doppler velocity log during prolonged (6-hour) operation.

grooves. The width and depth of the axial sealing grooves are 2.6 and 1.28 mm, respectively, the width and depth of the radial sealing grooves are 2.4 and 1.32 mm, respectively, and the wire diameter and outer diameter of the O-ring seal are 1.8 and 56 mm, respectively. In addition, a YMCBH6 water-tight connector connects the signal line from the top end of the sealed tank.

## Performance characterization
The transferred charges and current are measured by an electrometer (Keithley 6514, Tektronix, USA). The data are collected with a data acquisition card (NI USB-6008, National Instruments, USA) and the LabVIEW software platform. A home-designed ocean current simulation system is built to systematically characterize the DS-TENG's output performance. A diagram is shown in Supplementary Fig. 12. Resistors for impedance matching are calibrated using a precision impedance analyzer (E4980A, Keysight, USA).

## Deep-sea experiments
From May 2 to 4, 2023, we carried out a deep-sea validation of the DS-TENG in the waters near 110°46 'E and 16°73' N. The mother ship in this experiment was the 6000-tonne integrated research vessel TAN SUO ER HAO, designated TS2-24 for this voyage. A deep-sea experiment was carried out using the DS-TENG aboard an ROV. Two sea trials were conducted, with each sea test lasting ~ 6 h. The maximum test depth was 4531 m. The test process was as follows. (1) The ROV was kept stationary at depths of 100, 250, 500, 750, and 1000 m during its descent. At the same time, the DS-TENG obtained flow velocity information at different depths for a collection time of 10 s. (2) In the 4000-m abyssal ocean area, the ROV ran at a constant speed of 0.5, 1, 1.5, and 2 knots. During these runs, the DS-TENG continued its data collection routine for 10 s. This process was conducted twice for robust validation. (3) In the 4000-m abyssal ocean area, the ROV accelerated in the direction opposing the ocean current (with speed reaching a maximum within 20 s) and slowed after reaching the maximum speed (with the ROV power output decreasing to zero within 20 s), and the DS-TENG recorded and uploaded the data in real-time. This process was conducted twice. (4) Data were transmitted with a data acquisition frequency of 200 Hz and a transmission frequency of 20 Hz, and in RS-232 data communication mode.

## Data availability
The data that support the findings of this study are available from the corresponding authors upon request. Source data are provided with this paper.

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

## Acknowledgements

This work was financially supported by the National Key R&D Program of China (2021YFC3101300, C.Shang), National Natural Science Foundation of China (42222606, 62103400, 42211540003, 42381260314, 42376219, Y.Yang), Independent Project Deployed by Innovative Academy of Marine Information Technology of CAS (CXBS202103, Y.Yang), Strategic Priority Research Program (A) of the Chinese Academy of Sciences (XDA22040301, Development of seabed moving operation platform, M.Chen), Sanya Science and Technology Special Fund 2022KJCX66 (Y.Yang), 2024 Hainan International Science and Technology Cooperation Research and Development Project (GHYF2024013, H.Ma), and CAS Key Laboratory of Science and Technology on Operational Oceanography (No. OOST2021-07, C.Shang). The authors acknowledge Dr. Ning Yang and Ming Chen for their help with the sea expedition, the crews of the research vessel Tansuo 2, and Liwen Bianji (Edanz) (www.liwenbianji.cn) for editing the language of a draft of this manuscript.

## Author contributions

Y. Yang, C.B. Han, and C. Shang conceived the idea and analyzed the data. Y. Yang and Y.C. Pan wrote the paper. Z. Dai, H. Ma, and Y.C. Pan designed the structure of the DS-TENG system. C. Shang and Y. Yuan optimized the structure of the triboelectric nanogenerators. J. Zheng, J. Ling, C. Xie, W. Yang, Y. Yalikun, and X. Song helped with the experiments. All the authors discussed the results and commented on the manuscript.

## Competing interests

The authors declare no competing interests.
