## [Peer Review File · Nature Communications]

Self-powered and speed-adjustable sensor for abyssal ocean current measurements based on triboelectric nanogeneratorsREVIEWER COMMENTS

Reviewer #1 (Remarks to the Author):

This paper proposed a self-powered and speed-adjustable sensor for abyssal ocean-current measurements. The sensor shows some outstanding characteristics, such as fully integrated, self-powered, highly sensitive and withstand immense hydrostatic pressure. Impressively, this paper reported an underwater sensing experiment at depths down to 4531 m, showing a practical potential in the abyssal ocean. I read this paper with great interest. However, there are still some issues that need to be clarified. To be published in nature communications, following revisions must be conducted.

1. The author needs to explain how the DS-TENG can be "self-powered". And clearly defined it in the paper.
2. In Section Introduction, the author claimed that "The measurement range of the DS-TENG is 67% wider than the traditional vertical-axis current meter." However, the author only showed the maximum velocity of different flow velocity sensors in Fig. 4g and compared them with the sensor proposed in this paper. The minimum detected flow velocity of this paper also needs to be compared with that of the commercial sensors. Meanwhile, the wide flow range of the proposed sensor is achieved by two gears (i.e. transmission clearances), is it also necessary for traditional sensors?
3. Line 180: Is "rabbit fur" a natural animal material or a synthetic material? How durable is it? The loss after 30 days of operation needs to be quantitatively measured and demonstrated.
4. The DS-TENG can be driven by collecting ocean current with the rotating cup. This design used six conical cups symmetrically arranged around the center of a rotating disk. It seems that the design of rotating cup is directly related to the detected flow range of the sensor. At present, the author only mentioned the diameter of each rotating cup in Section Method, but does not show how they are arranged (e.g. diameter of the circular arrangement), so the author needs to draw the detailed design in Supplementary. In addition, it is mentioned in Method that the multiple blades were made using 3D printing. The performance of different rotating cups also needs to be shown in Supplementary, and explains why the rotating cup shown in the paper was finally selected.
5. What is the material used to constitute the sealing and tank? Does their thickness affect the maximum torque of the magnetic coupling? Please elaborate.
6. Line 210: The relationship between response time and flow velocity should be further clarified. Specifically, why claimed "the large transmission clearance at a low flow velocity"? A large transmission clearance corresponds to an extended response time, right? How to ensure the start-up sensitivity claimed by the author with the large transmission clearance? And how to balance response time and starting torque?
7. In Fig. 3b, the short-circuit current and rotational speeds also show a distinct correspondence. The reason for these changes needs to be analyzed and supplemented in the paper, such as that of output voltage and transferred charge in Line 225.
8. Line 234: The author claimed "the range of rotational speed is 0 ~ 1000", but there were only 8 frequency points to calibrate the speed in Fig. 3d. Please add more frequency measurement points.
9. Line 239: Why choose 120rpm as the lower speed to simulate the charging capacity under the deep-sea condition of a low flow velocity? It needs to be explained and supported by the literature or some other authoritative means.
10. Line 251: The mentioned pressure-resistance test lasted less than 20 minutes at 45 MPa. Is there any evidence or literature for this test and duration time? How to judge "the sensor remains intact", can the sensing test be repeated under high pressure?
11. Line 265: The author proposed that "the peak current may be affected by the external environment", but a sealed and pressure-resistant environment is provided, right? Please explain this seemingly superfluous concern and provide relevant literature. At the same time, it is proved that the selected voltage signal will not be affected.
12. Line 288: The measurement range of 0.02 to 6.69 m/s is realized by combining two transmission clearances. In the practical application of this paper, is the clearance pre-set by manual or can be real-time adjusted during the procedure? As the author claimed, this will affect the width of the measurement.
13. Writing needs improvement, existing some formatting problems. For example, the capacitance

size in 240 Line, the caption of Fig. 3d, etc.

14. Before Fig. 5, the voltage were tens of volts, but all of them were 3V in Fig. 5. The reason needs to be explained.

15. Is the circuit on the ROV in Fig. 5c the same as that in the sensor shown in Fig. 1a? The author needs to introduce these two circuits in more detail and show the circuit diagram.

16. It should be pointed out that the velocity shown in Fig. 5i refers to the ocean current velocity or the relative flow velocity between the ROV and seawater. Is it calibrated by the DVL?

Reviewer #2 (Remarks to the Author):

This paper presents a triboelectric nanogenerator (TENG) based self-powered ocean current measurement sensor. This system can distinguish the ocean current from 0.02 m/s to 6.69 m/s with systematic adaptation, which is a 67 % larger range than the commercial ocean current sensors. Along with this significant sensitivity, this system can be operated under the ocean even at a depth of 4500 m. In this regard, I think this paper has a distinct novel point to be published in Nature Communications after the revision process. Our comments to which the author can refer is suggested below:

1) Given that the magnetic coupling structure is utilized for the rotational energy transmission, the spacing between the two magnetic coupling can be easily adjusted, which allows the alternation of the sensing range. However, it seems that the spacing needs to be adjusted manually. Given that the author suggesting the TENG operation in deep sea, I think self-adjustment of the sensing range must be possible due to the low accessibility of the deep-sea environment. I think the author should consider about it for its novelty.

2) Also, I guess the magnet must be strong enough to affect each other. In this regard, if the electromagnetic generator is hybridized with the suggested TENG, it can be helpful to use this module as an energy harvesting source, not only for the ocean current sensing module. Given that energy harvesting on the inaccessible environment is always getting spotlight, it can enhance the applicability. The author should consider about it.

3) In Figure 2f, the transmission curve of the active and passive magnets at different transmission clearances is shown. However, it is hard to distinguish the data, because several data are duplicated. The author should refurbish the figure for the readers' understanding.

4) In Figures 4b to 4f, the electrical output generated from the two different state is shown. However, although the state 1 is well explained in Figure 4b, details about state 2 are not explained in the main figures. I think this figure set does not reveal the correlation between the operational state and the sensitivity range. In this regard, I think author should add about the detail about state 2 in Figure 4. It will enhance the readability of the figure.

5) In Figure 5e, the raw voltage output generated underwater is depicted. However, it seems that there are unclear data which can potentially decrease the accuracy of the current sensing. Where these abnormal peaks are coming from, and how these peaks will affect the sensing?

Reviewer #3 (Remarks to the Author):

This is a well-written manuscript and I think it is appropriate for this journal. The manuscript is well referenced with appropriate citations and is well illustrated with high quality figures. The design of the current meter has the potential to provide increased capabilities to the oceanographic community. I would recommend this paper for publication if these issues can be addressed:

1. It is not clear to me from the description in the paper that the power generated by the instrument is sufficient to power a self-contained acquisition and logging system, which would need to be integrated if the instrument were to be deployed as a stand-alone device that provides data on recovery from deployment. If this were the case then this would be a very significant advance as it would enable long-term monitoring.

2. I would like some clarification on this sentence: 'After 30 days of operation, the reduction in the induced charge of the TENG remains within 6%'. What is the cause of this reduction and would it continue beyond the 30 day period? What causes it and what effect does it have on the lifespan of the instrument? Maybe I have misinterpreted the sentence, but it would be good to have further details in the manuscript to ensure that others accurately interpret what is being said here.

In addition to these two points I have made various comments in an attached word document. I think the majority of these comments can be easily addressed and it would be good to see a revised manuscript.

Best wishes

**Institute of Deep-Sea Science and Engineering, Chinese Academy of Sciences
Luhuitou Road 28, Sanya, 572000 Hainan Province, People's Republic of China**

April 16th, 2024

RE: Manuscript resubmission to *Nature Communications*

Dear Reviewers:

Thank you for your valuable time in reviewing our manuscript “**A self-powered and speed-adjustable sensor for abyssal ocean-current measurements based on a nanogenerator**”. The original work (No.: NCOMMS-23-56530) has been reviewed for publication in *Nature Communications* and obtained constructive comments from the reviewers. To give a clear explanation, the comments and suggestions have been carefully considered and replied point-by-point. In the following, we quote the reviewers' reports in full, and our responses are interspersed in blue. Action taken is indicated in red.

Response to Reviewer #1:

Reviewer #1:

This paper proposed a self-powered and speed-adjustable sensor for abyssal ocean-current measurements. The sensor shows some outstanding characteristics, such as fully integrated, self-powered, highly sensitive and withstand immense hydrostatic pressure. Impressively, this paper reported an underwater sensing experiment at depths down to 4531 m, showing a practical potential in the abyssal ocean. I read this paper with great interest. However, there are still some issues that need to be clarified. To be published in nature communications, following revisions must be conducted.

Response: We greatly appreciate the generally positive feedback and thank the editors and reviewers for carefully reading our text and providing comments to improve our manuscript.

1. The author needs to explain how the DS-TENG can be “self-powered”. And clearly defined it in the paper.

Response: To elaborate the concept of the DS-TENG being “self-powered”, we begin by examining a conventional sensor. In general, conventional mechanical sensors used to measure the ocean current velocity have measurement, signal processing, and power supply modules. As the measurement module cannot generate electricity on its own, the power supply module must continuously power the measurement module throughout the measurement process. Taking the LS1206B mechanical flow meter as an example (Fig. R1a), in addition to the power consumed by the signal processing module, the generation of the pulsed electrical signal by the measurement module

Institute of Deep-Sea Science and Engineering, Chinese Academy of Sciences
 Luhuitou Road 28, Sanya, 572000 Hainan Province, People’s Republic of China

requires power to activate and deactivate the reed switch associated with the flow rate. Hence, both the measurement and signal processing modules rely on an external power supply.

Fig. R1. Schematic diagram of the working principles of (a) LS1206B and (b) the DS-TENG.

Unlike conventional sensors, the DS-TENG utilizes a triboelectric nanogenerator to self-generate power (without the need for an external power supply). It does this by generating electrical signals that correlate with the flow rate when triggered by water flow. These signals are then directly transmitted to the signal processing module. Although the signal processing module still requires a power supply, our work is an important development beyond the conventional sensor.

Moreover, by further enhancing the output power of the DS-TENG, the energy harvested by the TENG can be utilized to recharge the power supply module, thereby prolonging the device’s lifespan. Studies have widely reported that the advancement of TENG technology has facilitated the development of various self-powered sensors (Joule 2017, 1 (3), 480-521; Nat. Commun. 2020, 11 (1); Adv. Energy Mater. 2017, 7 (12); Nat. Commun. 2015, 6 (1)). Hence, our upcoming research will focus on optimizing the output performance of the TENG and developing efficient energy harvesting and management circuits to achieve fully self-powered operation of the current meter.

Action: The interpretation of the DS-TENG being “self-powered” has been clarified in the revised manuscript (Line 81).

2. In Section Introduction, the author claimed that “The measurement range of the DS-TENG is 67% wider than the traditional vertical-axis current meter.” However, the author only showed the maximum velocity of different flow velocity sensors in Fig. 4g and compared them with the sensor proposed in this paper. The minimum detected flow velocity of this paper also needs to be compared with that of the

**Institute of Deep-Sea Science and Engineering, Chinese Academy of Sciences
Luhuitou Road 28, Sanya, 572000 Hainan Province, People’s Republic of China**

commercial sensors. Meanwhile, the wide flow range of the proposed sensor is achieved by two gears (i.e. transmission clearances), is it also necessary for traditional sensors?

Response: We appreciate your suggestions. Indeed, the maximum and minimum velocity measurements should be compared simultaneously, as in Fig. 4g and Fig. R2. Here, our DS-TENG has a velocity measurement range of 0.02–6.67 m/s whereas a commercial mechanical current sensor (such as LS-45 or LB-70) has a measurement range of 0.01–4 m/s. Therefore, the measurement range of the DS-TENG is 67% wider than that of similar commercial products.

Limited by the fixed mechanical structure, conventional current meters are constrained to a specific measurement range. To obtain a broader velocity measurement range, the combination of multiple measurement ranges is necessary. Interestingly, the concept of multi-measurement range superposition is adopted in other instrumentation design. For instance, we consider an impedance analyzer like the Keysight E4980A, which offers five measurement ranges to measure impedance across a wide range of values through adjusting the gain multiplier of the vector ratio detector.

Employing this design strategy, we introduced the variable-transmission clearance structure, capable of switching between different measurement ranges. Our experiments demonstrated that this design strategy expanded the measurement range for ocean currents. Furthermore, we believe that the approach of achieving a wider velocity measurement range using the variable magnetic-coupling transmission clearances structure serves as a reference for the traditional flow velocity sensor.

Action: We have updated Fig. 4g to include both the maximum and minimum velocity measurement capabilities of the DS-TENG in comparison with similar commercial products.

Fig. R2. Comparison of the flow velocity measuring range between the DS-TENG and commercial current meters.

3. Line 180: Is “rabbit fur” a natural animal material or a synthetic material? How

**Institute of Deep-Sea Science and Engineering, Chinese Academy of Sciences
Luhuitou Road 28, Sanya, 572000 Hainan Province, People's Republic of China**

durable is it? The loss after 30 days of operation needs to be quantitatively measured and demonstrated.

Response: Rabbit fur is a natural animal material, which is extensively used in the clothing and various other industries after undergoing industrial standard treatments such as cleaning and bleaching to remove grease and impurities.

As rabbit fur exhibits excellent toughness and elasticity at the right length, using rabbit fur as the friction material is advantageous to the DS-TENG in terms of reducing material wear. Elastic and soft rabbit fur has been widely adopted as a triboelectric material (Pang, Hao, et al. "Segmented swing-structured fur-based triboelectric nanogenerator for harvesting blue energy toward marine environmental applications." *Advanced Functional Materials* 31.47 (2021): 2106398; Chen, Pengfei, et al. "Super-durable, low-wear, and high-performance fur-brush triboelectric nanogenerator for wind and water energy harvesting for smart agriculture." *Advanced Energy Materials* 11.9 (2021): 2003066; Han, Jiajia, et al. "Achieving a large driving force on triboelectric nanogenerator by wave-driven linkage mechanism for harvesting blue energy toward marine environment monitoring." *Advanced Energy Materials* 13.5 (2023): 2203219.). It has been reported that a wind-driven soft-contact rotary triboelectric nanogenerator based on rabbit fur demonstrated stable performance. Remarkably, the device exhibited no obvious wear even after 480,000 cycles (Han, Jiajia, et al. "Wind-driven soft-contact rotary triboelectric nanogenerator based on rabbit fur with high performance and durability for smart farming." *Advanced Functional Materials* 32.2 (2022): 2108580), showing that rabbit fur is effective in the simultaneous generation of high-density charges, reduction of frictional resistance, and extension of the lifetime of the device.

To quantitatively demonstrate the durability of the rabbit fur, after 30 days of use, we visually inspected the fur and found no noticeable wear on its surface. Furthermore, we used SEM to observe the surface topography of the rabbit fur before and after 30 days of use. As shown in Fig. R3, there is almost no wear on the fur surface, which indicates that the combination of rabbit fur and flexible polymer film is reasonable as the friction material of the TENG.

Fig. R3. SEM image of rabbit fur after running for more than 30 days.

Action: We have added SEM images of rabbit fur before and after 30 days of use in the Supplementary Information (Supplementary Fig. 3).

4. The DS-TENG can be driven by collecting ocean current with the rotating cup. This design used six conical cups symmetrically arranged around the center of a rotating disk. It seems that the design of rotating cup is directly related to the detected flow range of the sensor. At present, the author only mentioned the diameter of each rotating cup in Section Method, but does not show how they are arranged (e.g. diameter of the circular arrangement), so the author needs to draw the detailed design in Supplementary Information. In addition, it is mentioned in Method that the multiple blades were made using 3D printing. The performance of different rotating cups also needs to be shown in Supplementary Information, and explains why the rotating cup shown in the paper was finally selected.

Response: We thank the reviewer for highlighting the importance of the rotating cup design. The performance of the mechanical current meter hinges on several key factors, such as the inertia of the rotor, friction in the bearings, and the rotor's responsiveness to water flow (American Society of Civil Engineers New York, 1994: 376-385; Proceedings of the National Conference on Hydraulic Engineering. 1993 (pt 2): 1452-1457). As the mechanical current meter cup has a mature commercial design and usage, we opted to adhere directly the national standard of rotating-element current meters (GB/T11826-2019, China) in our design process. The standard dictates a range of three–six rotating cups with a rotational diameter between 50 and 150 mm. As previous studies have indicated that a current meter has an improved start-up performance when equipped with six rotating cups (Journal of Hydraulic Research, 2007, 45(6): 752-762), we opted for the standard six rotating cups and a moderate rotation diameter of 110 mm. The detailed design parameters of the DS-TENG

Institute of Deep-Sea Science and Engineering, Chinese Academy of Sciences
Luhuitou Road 28, Sanya, 572000 Hainan Province, People's Republic of China

rotating cup module are shown in Fig. R4. The opening diameter of the six rotating cups is 40 mm, and the length of the bus bar is 35 mm. The rotating cups are evenly distributed in the same plane and have a rotation diameter of 110 mm.

Action: We have explained the design principles and references of the rotating cups in the revised manuscript (Line 412) and included the detailed design of the DS-TENG rotating cup in the revised Supplementary Information (Supplementary Fig. 11).

Fig. R4. The detailed design parameters of the rotating cups of DS-TENG.

5. What is the material used to constitute the sealing and tank? Does their thickness affect the maximum torque of the magnetic coupling? Please elaborate.

Response: The pressure tank is made of 316 stainless steels, which is a high-strength austenitic stainless steel with high yield strength (310 MPa) and tensile strength (620 MPa). Therefore, this material is beneficial in terms of reducing the thickness of the isolation sleeve while meeting the requirements of deep-sea pressure resistance, and it is commonly used in deep-sea equipment (Oceans' 97. MTS/IEEE Conference Proceedings. IEEE, 1997, 2). In addition, the non-magnetic nature of 316 stainless steel ensures that its effect on the magnetic coupling is negligible.

The sealing end cover is designed on top of the DS-TENG, and the two grooves between the end cover and the pressure tank are sealed by an O-ring made of nitrile material, which is a commonly used sealing material in deep-sea equipment. Detailed information has been added to the materials section of the manuscript.

(1) The sealing structure is on the top of the DS-TENG and does not make contact with the magnetic coupling, such that it does not affect the transmission.

(2) The thickness of the sidewall of the pressure-resistant tank mainly affects the compressive strength and weight of the equipment and does not affect the magnetic transmission in this work, and we thus reduce the thickness of the sidewall as much as possible to reduce the weight while ensuring the compressive strength.

(3) For the thickness of the tank between the magnetic couplings, it is most important to consider both the pressure resistance and magnetic transmission

**Institute of Deep-Sea Science and Engineering, Chinese Academy of Sciences
Luhuitou Road 28, Sanya, 572000 Hainan Province, People's Republic of China**

performance. Theoretically, a smaller thickness corresponds to a smaller magnetic drive clearance. The magnetic drive clearance is negatively correlated with the torque transferred and positively correlated with the start-up time. In this work, experiments and calculations showed that the minimum thickness of the bottom of the tank that ensured sufficient pressure resistance in the deep sea was 11 mm.

Action: We have added details of the tank and sealing materials in the “Fabrication of the velocity sensor” section of the revised manuscript (Line 406).

6. Line 210: The relationship between response time and flow velocity should be further clarified. Specifically, why claimed “the large transmission clearance at a low flow velocity”? A large transmission clearance corresponds to an extended response time, right? How to ensure the start-up sensitivity claimed by the author with the large transmission clearance? And how to balance response time and starting torque?

Response: According to the transmission principle of the magnetic coupling, the two magnets of the magnetic coupling generate an axial force that attracts each magnet to the other in addition to transmitting torque. As shown in Fig. R5, the axial force increases the frictional resistance of the internal structure. In addition, as the transmission clearance decreases, the axial force gradually increases, resulting in greater internal frictional resistance and therefore worse start-up performance at low flow velocity. Therefore, at low flow velocity, we use a large clearance to reduce frictional resistance and thus achieve better start-up. Moreover, according to the acceleration formula $V_t = V_0 + at$ and $F = ma$, we have $F = mV_t/t$ when $V_0 = 0$ (where V_t is the final velocity, V_0 is the initial velocity, a is the acceleration, m is the mass of the object, and t is the time required to reach V_t). This indicates that the time required for the cup to reach the rotating speed (V_t) at a certain flow velocity is inversely proportional to the thrust of the flow (F). Therefore, owing to the small F at low flow velocity, a long response time is required. Indeed, a large transmission clearance corresponds to an extended response time.

Fig. R5. Diagram of the effect of the axial force of magnetic coupling on transmission resistance.

In this work, as the clearance increases from 12.5 to 18.5 mm, the response time increases from 0.48 to 0.96 s. Even though the use of a large transmission clearance at low flow velocity indeed sacrifices the response time to a certain extent, the response time is sufficient for long-term observations of upper-ocean flow (which varies over the course of hours or days) or abyssal flow (which varies over the course of months or seasons; Zhou et al., *Geophysical Research Letters* 49.3 (2022): e2021GL096530). Therefore, using a large transmission clearance fulfills the requirements for measuring slow currents with minimal variations in flow velocity.

Finally, we agree with the reviewer that it is important to balance the response time and starting torque. In the design phase, the response time of the DS-TENG was optimized by adjusting the transmission clearance, optimizing the length of rabbit fur, and selecting a bearing material in alignment with application requirements. To reduce the response time of the DS-TENG to a greater extent, on one hand, we can further optimize the material and structure of the TENG to reduce the friction brought by the TENG itself while ensuring sufficient signal output. On the other hand, the transmission clearance can be tailored for different use scenarios to meet the required response time and starting torque, and the transmission clearance can be minimized to reduce the response time under the condition of the minimum detection speed.

Action: We have included the above discussions in the Supplementary Information (Supplementary Fig. 5).

Institute of Deep-Sea Science and Engineering, Chinese Academy of Sciences
Luhuitou Road 28, Sanya, 572000 Hainan Province, People's Republic of China

7. In Fig. 3b, the short-circuit current and rotational speeds also show a distinct correspondence. The reason for these changes needs to be analyzed and supplemented in the paper, such as that of output voltage and transferred charge in Line 225.

Response: During the process of stable friction between the fur and fluorinated ethylene propylene (FEP), the triboelectric charge density remains constant over time.

The short-circuit current of the TENG (I_{sc}) is the instantaneous current and is calculated using Equation (1). I_{p-p} is the peak current of the TENG and is calculated using Equation (2). If the frequency of rotor rotation is f , the relationship between the charge transfer time (t) and (f) in the circuit is that given by Equation (3). From Eqs. (1) (2) (3), it is derived that I_{sc} is positively related to $1/f^2$ and I_{p-p} is positively related to f . As the rotational speed (n) is equal to the frequency of rotor rotation (f), the peak current I_{p-p} is positively related to the rotational speed (n).

$$I_{sc} = \frac{dQ}{dt} \quad (1)$$

$$I_{p-p} = \frac{dI_{sc}}{dt} \quad (2)$$

$$t = \frac{1}{4f} \quad (3)$$

Action: We have included a discussion in the revised manuscript (Line 241).

8. Line 234: The author claimed “the range of rotational speed is 0 ~ 1000”, but there were only 8 frequency points to calibrate the speed in Fig. 3d. Please add more frequency measurement points.

Response: Thank you for your comments. We added more frequency points in Fig. 3d. The original and revised figures are as follows.

Action: The curve in Fig. 3d of the manuscript has been replaced.

9. Line 239: Why choose 120 rpm as the lower speed to simulate the charging capacity under the deep-sea condition of a low flow velocity? It needs to be explained and supported by the literature or some other authoritative means.

Response: Thank you for your question. Here, the statement “To simulate the output capacity of the DS-TENG under the deep-sea condition of a low flow velocity” is inaccurate. We have revised the text to “For simulating the operational conditions of deep-sea exploration equipment at low speeds”. This correction is made because the DS-TENG is designed as a versatile flow velocity sensor applicable not only for measurement of the fixed-point flow rate (absolute flow velocity of seawater) but also for speed detection in underwater vehicles (such as autonomous underwater vehicles) during missions. Typically, autonomous underwater vehicles attain maximum speeds of 4 knots (approximately 2 m/s) whereas large submersibles (such as Orca XLUUV, USA) reach speeds exceeding 8 knots (approximately 4 m/s). Consequently, the speed range of the DS-TENG needs to cover the velocity range from the deep-sea flow velocity (a few centimeters per second) to the cruising speed of an underwater vehicle (a few meters per second). As the relative flow rate measured by the DS-TENG at 120 rpm is approximately 1 m/s, 120 rpm is selected as the lower speed in simulating the charging capability of the deep-sea exploration equipment at a certain low speed.

Action: Line 254: The text “To simulate the output capacity of the DS-TENG under the deep-sea condition of a low flow velocity” has been replaced with “For simulating the operating conditions of deep-sea exploration equipment at low speed”.

10. Line 251: The mentioned pressure-resistance test lasted less than 20 minutes at 45 MPa. Is there any evidence or literature for this test and duration time? How to judge “the sensor remains intact”, can the sensing test be repeated under high pressure?

Response: Thank you for your question. Most of the conditions in the hydrostatic pressure test adhere to Environmental Test Methods for Oceanographic Instruments – Part 15: Hydrostatic Pressure Test (GB/T 32065.15-2019, China). The pressurization rate, depressurization rate, and maximum test pressure were set in full accordance with the testing standards, and the only parameter that did not meet the standards was the duration of the test. Given our original plan to carry out a sea trial at a depth of 3600 m, we opted for a maximum test pressure of 45 MPa (calculated as 36 MPa × 1.25), paired with a compression rate and pressure reduction rate of 2 MPa/min. As mentioned, the selection of the pressure holding time (20 min) was not that of the standard. (According to the standard, the pressure holding time should be set within 1/2/4/8 h).

Institute of Deep-Sea Science and Engineering, Chinese Academy of Sciences
Luhuitou Road 28, Sanya, 572000 Hainan Province, People's Republic of China

In the actual deep-sea test, after completing the originally planned 3600-m trial, the DS-TENG's on-board platform (or remotely operated vehicle, ROV) temporarily conducted a deep-sea test at 4500 m. We were pleased to find that the DS-TENG continued to operate normally after more than 6 hours of testing at 4500 m. After the completion of the test, the DS-TENG remains operational, which indicates that the DS-TENG can be used repeatedly for sensing tests under high pressure. The judgment that “the sensor remains intact” is mainly based on the fact that the pressure tank had no deformation after the pressure test, the inside of the sealed tank remained dry, and the DS-TENG still worked normally.

Fig. R6. The original (left) and revised (right) hydrostatic pressure tests.

To better understand the repeatability and pressure resistance of the DS-TENG at 4500 m, we supplemented the standard pressure test of the DS-TENG (Fig. R6). With reference to the above standard (GB/T 32065.15-2019, China), the compression rate, pressure reduction rate and pressure holding time were set at 1 MPa/min, 2 MPa/min and 4 h respectively, and the number of cycles was set as two. The maximum test pressure was set at 49.5 MPa (considering the small size of the DS-TENG, the 4500 m test depth in the field, and an appropriate safety factor of 1.1 ($45 \text{ MPa} \times 1.1 = 49.5 \text{ MPa}$)). Our additional pressure test demonstrated that the DS-TENG maintained normal operation in repeated high-pressure tests. During the actual sea trial, the DS-TENG worked well in the two tests totaling 8 hours at 4500 m. In addition, we conducted two further deep-sea field tests totaling 8 hours, and the DS-TENG maintained normal operation after recovery, demonstrating that it performs stable operation under repeated high-pressure conditions.

Action: The curve in Fig. 3i of the manuscript has been replaced.

11. Line 265: The author proposed that “the peak current may be affected by the external environment”, but a sealed and pressure-resistant environment is provided, right? Please explain this seemingly superfluous concern and provide relevant literature. At the same time, it is proved that the selected voltage signal will not be

**Institute of Deep-Sea Science and Engineering, Chinese Academy of Sciences
Luhuitou Road 28, Sanya, 572000 Hainan Province, People's Republic of China**

affected.

Response: Thank you for your question. As you said, we have provided the TENG part with sealed and pressure-resistant conditions, such that the TENG has a stable working environment. The TENG is inside the sealed tank and the rotation of the TENG rotor is triggered by the water flow. Therefore, the “external environment” mentioned here refers more to different scenarios of the flow velocity.

According to the power generation principle of triboelectric nanogenerators, there is an obvious correspondence between the short-circuit current and the external trigger frequency (Nat. Commun. 2021, 12 (1)). For the TENG alone, according to $I = dQ/dt$ and $U = Q/C$, the voltage signal is relatively stable because the quantity Q is easily saturated and stabilized, and the capacitance C is relatively stable, whereas the current is heavily dependent on the rotational speed. As shown in Fig. 3b, a higher speed of the TENG rotor corresponds to greater peak current output. Therefore, in the case of low flow velocity, the short-circuit current of the TENG is low, and if the current is used as a sensing signal, the output waveform may be submerged by the presence of noise. In contrast, the peak voltage of the TENG output is not directly related to the external trigger frequency, which makes the output by the TENG stable and easy to parse in both low and high-flow velocity scenarios. Meanwhile, the frequency of voltage signals is positively correlated with the external trigger frequency, and we thus choose the voltage signal as the sensing signal.

Action: Line 284: We have corrected the text “the peak current may be affected by the external environment” to “the peak current may be affected by the external environment (for example, the rotating speed)”.

12. Line 288: The measurement range of 0.02 to 6.69 m/s is realized by combining two transmission clearances. In the practical application of this paper, is the clearance pre-set by manual or can be real-time adjusted during the procedure? As the author claimed, this will affect the width of the measurement.

Response: As you state, the velocity measurement interval of 0.02–6.69 m/s is achieved by combining the velocity measurement intervals of two transmission clearances. In the current work, this adjustment of the speed range is realized manually, by which the speed range is increased. The flexible design of this variable clearance structure allows for further increases in the minimum and maximum flow velocities by optimizing the clearance and a further setup of multiple gears to handle more flow velocity test scenarios.

Therefore, in subsequent work, we will realize the automatic mechanical adjustment of the speed measurement interval by designing the scheme shown in Fig.

Institute of Deep-Sea Science and Engineering, Chinese Academy of Sciences
Luhuitou Road 28, Sanya, 572000 Hainan Province, People's Republic of China

R7. By designing an additional rotor, a chute-iron ball mechanism is integrated into the blades of the rotor, with the iron ball connected via steel wire and a housing (green part) that controls the clearance of the magnetic coupling. Our structure is based on the centrifugal force generated during the rotation of the small iron ball to generate tension in the axial direction, which moves the active magnet upward. In this way, the mechanical regulation of the transmission clearance by the rotational speed can be realized. Specifically, in the initial state, the DS-TENG defaults to the low flow velocity with the highest accuracy (when the transmission clearance is a maximum, Fig. R7a). When the rotating speed of the rotor increases with the flow velocity, the iron ball moves outward under the action of the centrifugal force and pulls the green part upward (Fig. R7b), such that the transmission clearance is reduced. Thus, the measurement range can be real-time adjusted with this structure.

Fig. R7. Schematic diagram of automatic adjustment system for velocity measuring interval. (a) Small flow conditions and (b) large flow conditions.

13. Writing needs improvement, existing some formatting problems. For example, the capacitance size in 240 Line, the caption of Fig. 3d, etc.

Response: Thank you for your suggestion. We have made corresponding formatting changes in the revised manuscript.

14. Before Fig. 5, the voltage were tens of volts, but all of them were 3 V in Fig. 5. The reason needs to be explained.

Response: The direct output voltage (tens of volts) of the TENG is beyond the input voltage of the analog-to-digital converter. In safeguarding the signal processing circuit,

**Institute of Deep-Sea Science and Engineering, Chinese Academy of Sciences
Luhuitou Road 28, Sanya, 572000 Hainan Province, People's Republic of China**

the output voltage of the TENG is reduced to several volts by circuit divides (10:1) to ensure that the input signal is within the acceptable range. Consequently, the voltages in Fig. 5 are approximately 3 V.

15. Is the circuit on the ROV in Fig. 5c the same as that in the sensor shown in Fig. 1a? The author needs to introduce these two circuits in more detail and show the circuit diagram.

Response: The circuit for the ROV in Fig. 5c is the same as that in Fig. 1a. As shown in Fig. R8, the original signal generated by the TENG is input into the circuit board, which includes an acquisition, processing, and communication circuit. The function of the acquisition circuit is to acquire the high-resistance and high-voltage signals of the TENG. First, the acquisition circuit divides and filters the TENG signals. Then, the microcontroller in the processing circuit performs analog-to-digital conversion and data encoding of analog signals through an analog-to-digital converter. Finally, the communications circuit further transmits the encapsulated data to the ground platform via the ROV platform.

In addition, to avoid a misunderstanding, we modified Fig. 5c such that it is consistent with the schematic in Fig. 1c. The modified figure is shown in Fig. R9.

Fig. R8. The circuit diagram of the DS-TENG.

The original (top) and revised (bottom) images are as follows.

Fig. R9. Corrected diagram of Fig. 5c

Action: We corrected Fig. 5c in the revised manuscript. Furthermore, the details of the circuit design are explained and included in the Supplementary Information (Supplementary Fig. 9).

16. It should be pointed out that the velocity shown in Fig. 5i refers to the ocean current velocity or the relative flow velocity between the ROV and seawater. Is it calibrated by the DVL?

Response: The velocity shown in Fig. 5i refers to the relative flow velocity between the ROV and seawater, which is calibrated by the DVL.

Action: Line 346: The text “this robust response to small velocity changes under abyssal zone conditions is supported by the linear correlation between the different flow velocities” has been revised to “this robust response to small velocity changes under abyssal zone conditions is supported by the linear correlation between the different relative flow velocities (between the ROV and seawater)”.

**Institute of Deep-Sea Science and Engineering, Chinese Academy of Sciences
Luhuitou Road 28, Sanya, 572000 Hainan Province, People's Republic of China**

Response to Reviewer #2:

This paper presents a triboelectric nanogenerator (TENG) based self-powered ocean current measurement sensor. This system can distinguish the ocean current from 0.02 m/s to 6.69 m/s with systematic adaptation, which is a 67 % larger range than the commercial ocean current sensors. Along with this significant sensitivity, this system can be operated under the ocean even at a depth of 4500 m. In this regard, I think this paper has a distinct novel point to be published in Nature Communications after the revision process. Our comments to which the author can refer is suggested below:

Response: We greatly appreciate the positive feedback and thank the reviewer again for carefully reviewing our manuscript.

1) Given that the magnetic coupling structure is utilized for the rotational energy transmission, the spacing between the two magnetic coupling can be easily adjusted, which allows the alternation of the sensing range. However, it seems that the spacing needs to be adjusted manually. Given that the author suggesting the TENG operation in deep sea, I think self-adjustment of the sensing range must be possible due to the low accessibility of the deep-sea environment. I think the author should consider about it for its novelty.

Response: We strongly agree with the reviewer. Owing to the poor reachability in a deep-sea environment, it is necessary (and possible) to realize the self-adjustment of the sensing range when the actual flow velocity is out of the current range. For example, we designed a structure (Fig. R10) to realize the adjustments of axial spacing. This structure uses the centrifugal force generated by removable iron ball in blades during the rotation of the rotors to form an axial tension that moves the active magnet upward. In this way, the mechanical regulation of the transmission clearance by the rotational speed can be realized. Specifically, in the initial state, the DS-TENG defaults to the low flow velocity with the highest accuracy (when the transmission clearance is a maximum, Fig. R10a). When the rotating speed of the rotor increases with the flow velocity, the iron ball moves outward under the action of the centrifugal force and pulls the green part upward (Fig. R10b), such that the transmission clearance is reduced. Thus, the measurement range can be real-time adjusted with this structure.

Fig. R10. Schematic diagram of automatic adjustment system for velocity measuring interval. (a) Small flow conditions and (b) large flow conditions.

2) Also, I guess the magnet must be strong enough to affect each other. In this regard, if the electromagnetic generator is hybridized with the suggested TENG, it can be helpful to use this module as an energy harvesting source, not only for the ocean current sensing module. Given that energy harvesting on the inaccessible environment is always getting spotlight, it can enhance the applicability. The author should consider about it.

Response: Thank you for your suggestion. The TENG has the advantage of high voltage and the electromagnetic generator (EMG) has the advantage of high current. Coupling an EMG and TENG can improve the energy harvesting efficiency and output power of the whole system. In addition, the energy harvesting technology for marine environments based on the coupling of the TENG and EMG has been widely studied (Adv. Energy Mater. 2021, 11(31): 2101116; Adv. Electron. Mater. 2021, 7(9): 2100277; Small 2023, 19(25): 2300847), and we will further consider how to integrate an EMG inside the DS-TENG in future work to improve the environmental adaptability of the DS-TENG.

3) In Figure 2f, the transmission curve of the active and passive magnets at different transmission clearances is shown. However, it is hard to distinguish the data, because several data are duplicated. The author should refurbish the figure for the readers' understanding.

Response: Thank you for the suggestion. Owing to a data overlap in Fig. 2f, detailed data curves (Fig. R11) are presented in the Supplementary Information.

The figure in manuscript is as follows.

The figure in the Supplementary Information is as follows.

Fig. R11. Transmission curve at different transmission clearances. (a) 0 mm, (b) 12.5 mm, (c) 15.5 mm, (d) 18.5 mm, (e) 21.5 mm, (f) 24.5 mm.

Action: We have added Fig. R11 to the Supplementary Information (Supplementary Fig. 4).

4) In Figures 4b to 4f, the electrical output generated from the two different state is shown. However, although the state 1 is well explained in Figure 4b, details about state 2 are not explained in the main figures. I think this figure set does not reveal the correlation between the operational state and the sensitivity range. In this regard, I think author should add about the detail about state 2 in Figure 4. It will enhance the readability of the figure.

Response: Thank you for your suggestion. We recognize that not showing the two states might prevent the readers from understanding that the two states correspond to the two sensitivities.

Editorial Note: Parts of the figure below have been redacted as indicated to remove third-party material where no permission to publish could be obtained.

Institute of Deep-Sea Science and Engineering, Chinese Academy of Sciences
 Luhuitou Road 28, Sanya, 572000 Hainan Province, People's Republic of China

Action: Figure 4 has been revised to include state 2 and thus enhance the reader's comprehension of the distinction between states 1 and 2.

The original (top) and revised (bottom) images are as follows.

**Institute of Deep-Sea Science and Engineering, Chinese Academy of Sciences
Luhuitou Road 28, Sanya, 572000 Hainan Province, People's Republic of China**

5) In Figure 5e, the raw voltage output generated underwater is depicted. However, it seems that there are unclear data which can potentially decrease the accuracy of the current sensing. Where these abnormal peaks are coming from, and how these peaks will affect the sensing?

Response: Thank you for your question. Measurement circuits for voltage can be categorized into impedance-type circuits (e.g., oscilloscopes) and capacitive-type circuits (e.g., electrostatic meters). In our circuit design, the impedance-type circuit design is used. When the rotor of the DS-TENG rotates slowly or does not rotate, as the TENG can be modeled as a capacitor, the impedance-type circuit generates abnormal peaks due to the generation of electrical current during the voltage acquisition. Therefore, to avoid this situation, future improvements of the signal processing circuit need to consider the matching of impedance and capacitance characteristics, and the combination of the two to realize a better signal processing method.

Response to Reviewer #3:

This is a well-written manuscript and I think it is appropriate for this journal. The manuscript is well referenced with appropriate citations and is well illustrated with high quality figures. The design of the current meter has the potential to provide increased capabilities to the oceanographic community. I would recommend this paper for publication if these issues can be addressed:

Response: We sincerely thank the reviewer for carefully reviewing our manuscript and making professional comments that are important to improving the quality of the manuscript.

1. It is not clear to me from the description in the paper that the power generated by the instrument is sufficient to power a self-contained acquisition and logging system, which would need to be integrated if the instrument were to be deployed as a stand-alone device that provides data on recovery from deployment. If this were the case then this would be a very significant advance as it would enable long-term monitoring.

Response: We appreciate and agree with your views. In general, conventional mechanical sensors used to measure the ocean current velocity have measurement, signal processing, and power supply modules. As the measurement module cannot generate electricity on its own, the power supply module must continuously power the measurement module throughout the measurement process (Fig. R12a). Hence, both the measurement and signal processing modules rely on an external power supply.

Unlike conventional sensors, the DS-TENG utilizes a triboelectric nanogenerator

**Institute of Deep-Sea Science and Engineering, Chinese Academy of Sciences
Luhuitou Road 28, Sanya, 572000 Hainan Province, People’s Republic of China**

to self-generate power (without the need for an external power supply, Fig. R12b) by generating electrical signals correlated to the flow rate when triggered by water flow. These signals are then directly transmitted to the signal processing module.

At present, the power generated by the DS-TENG under conditions of low flow velocity is approximately 30 μ W, while the signal processing and storage module are powered by a 3-V DC power (Li-ion battery) with an operating current of 3 mA. Thus, the DS-TENG in its present state is insufficient to power a self-contained acquisition and logging system as a stand-alone device. Although the signal processing module still requires a power supply, our work is an important development beyond the conventional sensor.

To realize the self-supply of the entire flow velocity measurement system and enable the long-term monitoring of the deep-sea flow velocity, improvements can be made: (1) optimizing the friction material of the TENG to increase surface charge density for improved output power; (2) enhancing the rotational and friction grid structure of the TENG to expand the effective friction area and optimize the induced charge flow velocity, thereby boosting the output voltage and current; and (3) refining the low-power signal processing circuit and strategizing the division of tasks between periodic signal acquisition and electrical energy harvesting to reduce the overall system power consumption (Fig. R12c). Through the above improvements and optimization, we expect to realize an ocean current sensor with a self-contained acquisition and logging system for long-term monitoring.

Fig. R12. Schematic diagram of the working principles of current sensors: (a) traditional system, (b) present DS-TENG, and (c) next DS-TENG.

2. I would like some clarification on this sentence: “After 30 days of operation, the reduction in the induced charge of the TENG remains within 6%”. What is the cause of this reduction and would it continue beyond the 30 days period? What causes it and what effect does it have on the lifespan of the instrument? Maybe I have misinterpreted the sentence, but it would be good to have further details in the manuscript to ensure that others accurately interpret what is being said here.

Response: The output performance of the TENG is affected by external environmental factors and TENG's inherent factors. Environmental factors primarily comprise temperature, humidity, radiation, and electromagnetic interference. This manuscript does not consider external environmental factors (e.g., humidity, pressure, and corrosiveness) because the TENG is isolated in a sealed tank.

TENG's inherent factors are described as follows.

(1) Material and wear factors: First, the two friction materials (FEP and fur) have good chemical stability such that they will generally last for many years. Second, the materials have good abrasive resistance. In principle, during the long-term friction process, the two materials rubbing against each other leave material debris on one another, which may reduce the surface charge density of the tribomaterials and thereby affect the TENG output. Given that the rabbit fur and FEP used in this study have high wear resistance and the contact friction between them is slight, the material and wear factors do not greatly affect the material lifetimes.

(2) Structural factors: The amount of triboelectric charge is related to the degree of contact between the two friction materials. The rabbit fur used in this study makes good contact with the FEP in the initial stage. With an increase in the running time, the contact force between the rabbit fur and FEP decreases because of the poor elastic restoring force, which reduces the area of contact and thus attenuates the output charge. Theoretically, the lifetime of the TENG might be over 10 years if the performance decay due to contact is avoided. Therefore, to increase the lifetime of the DS-TENG, the rabbit fur structure can be further optimized (1) by surface treatment of the rabbit fur, or by further reducing the fur length to increase the fur's elasticity for long-term stable contact with the FEP, or (2) by replacing the rabbit fur with other polymer brush structures with better elasticity to ensure a stable contact in the long-term.

In this work, the cause of the slight reduction in the TENG performance beyond the 30-day period was mainly attributed to the TENG's inherent factors (i.e., material wear or contact deformation of the rabbit fur). Nevertheless, the deformation of the rabbit fur and the attenuation of the induced charge will slow down, and the output of the TENG tends to fluctuate within in a certain range. In quantitatively demonstrating the durability of the rabbit fur, through visual inspections and SEM observations of the rabbit fur after 30 days of use (Fig. R12), we found no notable wear on the surface, indicating that it is appropriate to combine rabbit fur and FEP as the tribomaterials of the TENG. Furthermore, given that the sensor data are derived from the frequency of the output voltage rather than the amplitude, the signal-to-noise ratio remains substantial even if the charge continues to attenuate following the observed trend for over a year. Consequently, we believe that the reduction in charge would have

Institute of Deep-Sea Science and Engineering, Chinese Academy of Sciences
Luhuitou Road 28, Sanya, 572000 Hainan Province, People's Republic of China

minimal impact on the lifespan of the instrument. In future work, the lifespan of the DS-TENG will be further improved by optimizing the materials and structure.

Fig. R12. SEM image of rabbit fur after running for more than 30 days.

Action: We have revised the manuscript (Line 188) and included a detailed discussion in the Supplementary Information (Supplementary Fig. 3).

Once again, the authors would like to thank the reviewers for their constructive comments concerning our manuscript entitled “**A self-powered and speed-adjustable sensor for abyssal ocean-current measurements based on a nanogenerator**” (No.: NCOMMS-23-56530).

Sincerely yours,

Yang Yang, Prof. Dr.
Institute of Deep-Sea Science and Engineering
Chinese Academy of Sciences
Luhuitou Road 28, Sanya, Hainan 572000
P.R. China

Chang Bao Han, Prof. Dr.
Faculty of Materials and Manufacturing
Beijing University of Technology
Pingleyuan 100, Chaoyang, Beijing 100124
P.R. China

REVIEWER COMMENTS

Reviewer #1 (Remarks to the Author):

All concerns from the reviewer have been addressed well, this manuscript is recommended to be published on Nature Communications in current version.

Reviewer #2 (Remarks to the Author):

First of all, I appreciate of the authors' efforts to improve the quality of the manuscript. The author kindly responded to the comments which are given. With the responses, the whole manuscript becomes more straightforward for readers to understand. So, I think this paper can be published on nature communications.

Reviewer #3 (Remarks to the Author):

Thank you for you comprehensive responses to myself and the other two reviewers. I would be happy for the paper to be published if further consideration could be given to clearly explaining what is meant by self-powered. This point was also raised by Reviewer 1 and I don't think the amendments made (Line 81) sufficiently address this point to fully explain to the reader. This is critical to be clear here as it is in the title of the paper.

**Institute of Deep-Sea Science and Engineering, Chinese Academy of Sciences
Luhuitou Road 28, Sanya, 572000 Hainan Province, People's Republic of China**

May 16th, 2024

RE: Manuscript resubmission to *Nature Communications*

Dear Reviewers:

Thank you for your valuable time in reviewing our manuscript “**A self-powered and speed-adjustable sensor for abyssal ocean-current measurements based on a nanogenerator**”. We appreciate your consideration of our revisions to the manuscript (No.: NCOMMS-23-56530A). Here, we have further revised the manuscript in response to your comments. In the following, we quote the reviewers' reports in full, and our responses are interspersed in **blue**. Action taken is indicated in **red**.

Response to Reviewer #1:

Reviewer #1:

All concerns from the reviewer have been addressed well, this manuscript is recommended to be published on Nature Communications in current version.

Response: We greatly appreciate the positive feedback and thank the reviewer for carefully reading our text and providing comments to improve our manuscript.

Response to Reviewer #2:

Reviewer #2:

First of all, I appreciate of the authors' efforts to improve the quality of the manuscript. The author kindly responded to the comments which are given. With the responses, the whole manuscript becomes more straightforward for readers to understand. So, I think this paper can be published on nature communications.

Response: We greatly appreciate the positive feedback and thank the reviewer again for carefully reviewing our manuscript.

Response to Reviewer #3:

Reviewer #3:

Thank you for you comprehensive responses to myself and the other two reviewers. I would be happy for the paper to be published if further consideration could be given to clearly explaining what is meant by self-powered. This point was also raised by Reviewer 1 and I don't think the amendments made (Line 81) sufficiently address this point to fully explain to the reader. This is critical to be clear here as it is in the title of the paper.

**Institute of Deep-Sea Science and Engineering, Chinese Academy of Sciences
Luhuitou Road 28, Sanya, 572000 Hainan Province, People's Republic of China**

Response: We sincerely thank the reviewer for carefully reviewing our manuscript and making professional comments that are important to improving the quality of the manuscript.

To further explain what is meant by the term “self-powered”, we have added a description of the definition of “self-powered” in the Introduction of the revised manuscript (Line 67), with references to the relevant literature (Nat. Commun. 2020, 6, 1-8; Joule 2017, 1, 480-521; Nat. Rev. Cardiol. 2020, 18, 7-21; Sci. Adv. 2017, 3, 1-7; Sci. Adv. 2020, 6, 1-10).

Action: “For self-powered sensors, the transform from mechanical signals to recognizable electrical sensing signals does not need an additional power supply, and the TENG can perform this function well. For example, TENG-based self-powered sensing has been extensively studied in speed, vibration, biomedicine and other sensing fields” was added to the Introduction of the manuscript (Line 67), with the addition of the corresponding references (References 31-35).

Once again, the authors would like to thank the reviewers for their constructive comments concerning our manuscript entitled “**A self-powered and speed-adjustable sensor for abyssal ocean-current measurements based on a nanogenerator**” (No.: NCOMMS-23-56530A).

Sincerely yours,

Yang Yang, Prof. Dr.
Institute of Deep-Sea Science and Engineering
Chinese Academy of Sciences
Luhuitou Road 28, Sanya, Hainan 572000
P.R. China

Chang Bao Han, Prof. Dr.
Faculty of Materials and Manufacturing
Beijing University of Technology
Pingleyuan 100, Chaoyang, Beijing 100124
P.R. China

REVIEWERS' COMMENTS

Reviewer #3 (Remarks to the Author):

Thank you for the modifications. I would be happy for the manuscript to be published in its current form.